# RanBP1: A Potential Therapeutic Target for Cancer Stem Cells in Lung Cancer and Glioma

**DOI:** 10.3390/ijms24076855

**Published:** 2023-04-06

**Authors:** Yeon-Jee Kahm, In-Gyu Kim, Rae-Kwon Kim

**Affiliations:** 1Department of Radiation Biology, Environmental Safety Assessment Research Division, Korea Atomic Energy Research Institute, Yuseong-gu, Daejeon 34057, Republic of Korea; 2Department of Radiation Science and Technology, Korea University of Science and Technology, Yuseong-gu, Daejeon 34113, Republic of Korea

**Keywords:** RanBP1, CSC, EMT, IL-18

## Abstract

Cancer stem cells (CSCs) are known to be one of the factors that make cancer treatment difficult. Many researchers are thus conducting research to efficiently destroy CSCs. Therefore, we sought to suggest a new target that can efficiently suppress CSCs. In this study, we observed a high expression of Ran-binding protein 1 (RanBP1) in lung cancer stem cells (LCSCs) and glioma stem cells (GSCs). Upregulated RanBP1 expression is strongly associated with the expression of CSC marker proteins and CSC regulators. In addition, an elevated RanBP1 expression is strongly associated with a poor patient prognosis. CSCs have the ability to resist radiation, and RanBP1 regulates this ability. RanBP1 also affects the metastasis-associated epithelial–mesenchymal transition (EMT) phenomenon. EMT marker proteins and regulatory proteins are affected by RanBP1 expression, and cell motility was regulated according to RanBP1 expression. The cancer microenvironment influences cancer growth, metastasis, and cancer treatment. RanBP1 can modulate the cancer microenvironment by regulating the cytokine IL-18. Secreted IL-18 acts on cancer cells and promotes cancer malignancy. Our results reveal, for the first time, that RanBP1 is an important regulator in LCSCs and GSCs, suggesting that it holds potential for use as a potential therapeutic target.

## 1. Introduction

Ran binding protein 1 (RanBP1), which is expressed between eukaryotes and vertebrates, but not in some invertebrate species such as flies and worms, is a Ran-GTP-binding protein [1]. It is a co-activator of the Ran-GTPase activating protein, RanGAP1 [2]. RanBP1 plays a role in Ran-dependent nucleocytoplasmic transport [3]. Various studies have been performed on Ras-related nuclear protein (Ran) and it is overexpressed in various cancers [4,5,6,7]. Ran is regulated by Ran-GTPase and Ran-GDPase, and binds to various proteins including RanBP1 to transport Ran to the nucleus or regulate protein stabilization [8]. RanBP1 can form a stable complex with Ran and chromosome condensation regulator (RCC1), but its function is not clearly known. It also promotes the dissociation of Ran in complexes with KPNA2 and CSE1L [9]. RanBP1 may be the key to regulating Ran.

The regulation of secreted factors by RanBP1 is unknown. In this study, it was confirmed that IL-18 is regulated by RanBP1, and that cancer stem cells (CSCs) and the epithelial−mesenchymal transition (EMT) phenomenon are regulated by *IL-18*. The *IL-18* gene encodes a 193 amino acid protein [10]. The mature IL-18 secretory protein has a molecular weight of approximately 17.2 kDa [11]. IL-18 was initially known as an IFN-γ inducing factor [10,12]. This is because IFN-γ was enhanced after the injection of IL-18 and bacterial lipopolysaccharide (LPS) into mice [13]. IL-18 is a member of the IL-1 family [14]. The IL-1 family consists of seven cytokines with a proinflammatory activity, one anti-inflammatory cytokine, and three receptor antagonists [15,16]. The seven proinflammatory active cytokines are IL-1α, IL-1β, IL-18, IL-33, IL-36α, IL-36β, and IL-36γ. The anti-inflammatory cytokine is IL-37. Finally, the three receptor antagonists are IL-1RA, IL-36Rα, and IL-38 [17]. Cancer is characterized by tumor promoting inflammation and blocking immunity [18]. The pro-inflammatory cytokine IL-18 modulates the immune microenvironment in multiple myeloma and suppresses the action of T cells [19,20].

In this study, IL-18 was found to be regulated by RanBP1 in lung cancer stem cell (LCSC) and glioma stem cell (GSC), and CSC characteristics and EMT were regulated through this signaling mechanism.

## 2. Results

### 2.1. RanBP1 Expressed in Malignant Lung Cancer Cells

We identified genes that are overexpressed in ALDH1^+^ lung cancer cells from previous studies [21]. We obtained a list of genes overexpressed in CD133^+^ glioma from another research group [22]. We found a common overexpressed gene, *RanBP1*, in both gene lists. Through previous studies, it was found that A549 cells, a lung cancer cell line, have a high expression of ALDH1, a CSC marker protein [21]. On the other hand, it was confirmed that the expression of ALDH1 was low in H460 cells [23]. Therefore, we analyzed the expression of CSC marker proteins and *RanBP1* in A549 and H460 cells. Figure 1A shows the difference in the gene expression of CSC markers and *RanBP1* according to qPCR results. As a result, the gene expression of the CSC marker genes *ALDH1A1*, *ALDH1A3*, *CD133*, and *RanBP1* was higher in A549 cells than in H460. To confirm the difference in the protein expression level, the difference between CSC marker protein and RanBP1 expressed in A549 cells and H460 cells was analyzed (Figure 1B). Similar to gene expression, protein expression was also high in the A549 cells. Using the immunocytochemistry (ICC) experimental technique, differences in the expression of the CSC marker proteins and RanBP1 were visually compared and analyzed (Figure 1C). The expression of ALDH1A1 and CD133, which are CSC marker proteins, was high in A549 cells, and the expression of RanBP1 was also high in A549 cells.

### 2.2. Lung CSCs Regulated by the Expression of RanBP1

CSCs have the ability to form spheres under tumorsphere medium (TM) conditions. The sphere formation ability was measured according to the expression level of RanBP1 (Figure 2A). As a result, the formation of spheres was remarkably reduced in the group in which the *RanBP1* gene was suppressed. Figure 2B presents the results of a single cell assay that was performed to confirm the self-renewal ability of CSCs. When the *RanBP1* gene was suppressed using si-RNA, the sphere formation rate in single cells was significantly reduced. As another experiment to confirm the self-renewal ability of CSCs, a limited dilution assay was conducted (Figure 2C). As a result, the formation rate of spheres was significantly reduced in the si-RanBP1 group, identical to the results of the single cell assay. Figure 2D compares the expression levels of CD44, ALDH1A1, and ALDH1A3, which are marker proteins of LCSCs, by Western blot. It was confirmed that marker proteins of LCSCs were decreased in the group in which the RanBP1 expression was decreased. Expressions of SOX2, Oct-4, and Nanog, which are regulatory proteins that control CSCs, were also compared (Figure 2E). As a result, the expression of the regulatory proteins was decreased in the group with a low expression of RanBP1, consistent with the result for the marker proteins of LCSCs. A comparison of the expression levels of marker proteins in LCSCs using the ICC assay revealed that the expression of marker proteins ALDH1A1, ALDH1A3, and CD44 decreased in the group with a low RanBP1 expression (Figure 2F). CSCs have resistance to radiation, which makes radiation treatment difficult. A colony forming assay was thus conducted to evaluate the radioresistance ability according to the expression of RanBP1 (Figure 2G). The radiation dose used was 3Gy, showing a colony formation rate of about 50% compared with the control group. On the other hand, the group irradiated after suppressing the *RanBP1* gene showed approximately a 20% colony formation rate compared with the control group.

### 2.3. RanBP1 Involved in Regulating EMT Phenomenon

It is known that the EMT phenomenon appears in CSCs [24,25]. CSCs are known to be the main cause of metastasis, and EMT is the first process that must take place for metastasis to occur [26]. In order to confirm that the EMT phenomenon is regulated by RanBP1, an experiment was conducted after suppressing the gene expression of *RanBP1* using si-RNA. In the group in which the expression of RanBP1 was suppressed, the expression of epithelial marker protein, E-cadherin, increased, and the expression of mesenchymal marker proteins, N-cadherin and Vimentin, decreased (Figure 3A). The expression of Snail, Slug, Twist, and Zeb1, which are EMT-regulating proteins, was also decreased in cells with a low *RanBP1* gene expression (Figure 3B). Figure 3C was used to visually analyze the expression levels of EMT marker proteins using ICC. It can be seen that the expression of E-cadherin increased, and the expression of N-cadherin and Vimentin decreased, as shown in Figure 3A. One of the most significant features of the EMT phenomenon is an increase in cell motility [27]. Therefore, cell motility was observed using a Boyden chamber to measure cell migration and invasion (Figure 3D). In the group in which the expression of RanBP1 was lowered by treatment with si-RNA, cell migration and invasion ability were dramatically reduced. Additional experiments were conducted to measure the migration ability of the cells. The cell migration ability was measured by a wound healing assay, and the cell migration ability was significantly reduced in the si-RanBP1 group (Figure 3E).

### 2.4. Identification of Cytokines Regulated by RanBP1 and Their Impact on LCSCs and EMT

Cancer is self-regulated by cytokines and creates a favorable environment by adjusting the microenvironment [28,29]. There have been no studies on cytokines regulated by *RanBP1*. Thus, experiments were conducted to confirm cytokines regulated by *RanBP1* (Figure 4A). As a result of the cytokine array, the factors reduced by si-RanBP1 were CXCL1, IL-8, and IL-18. The RNA expression was compared when the expression of RanBP1 was inhibited by treatment with si-RNA (Figure 4B). The results showed that *CXCL1*, *IL-8*, and *IL-18* were decreased in the same manner as the cytokine array results. The protein levels of CXCL1, IL8, and IL-18 were reduced in the si-*RanBP1*-treated group (Figure 4C). The effect of each cytokine on LCSC and EMT was confirmed by treatment with a neutralizing antibody. It was confirmed that sphere forming was regulated in the antibody-treated group (Figure 4D). The greatest decrease was seen in the group treated with the IL-18 antibody. Figure 4E shows the motility of cells after the treatment with antibodies. As a result, cell migration and invasion were most inhibited in the IL-18 antibody-treated group, as shown in Figure 4D. Taken together, the results show that IL-18 had the strongest influence on the above results. After treatment with the IL-18 neutralizing antibody, we assessed whether CSC marker proteins were regulated (Figure 4F). It was found that the CSC marker proteins CD44, ALDH1A1, and ALDH1A3 were regulated in the antibody-treated group. Figure 4G shows that EMT marker proteins decreased after treatment with the IL-18 neutralizing antibody. To examine whether an autocrine loop was formed by IL-18 regulated by RanBP1, the expression of RanBP1 was observed after treatment with an IL-18 neutralizing antibody (Figure 4H). It was found that RanBP1 was regulated by IL-18, indicating that RanBP1 and IL-18 form a signaling loop.

### 2.5. Regulation of GSCs by RanBP1 Expression

*RanBP1* is a gene highly expressed in both LCSCs [21] and GSCs [22]. Therefore, an experiment was conducted to determine whether RanBP1 regulates GSCs. U87 cells, a glioma cell line, were used and tested by treating with TM. First, when the expression of RanBP1 was suppressed in U87 cells, GSC marker proteins were also suppressed (Figure 5A). As a result of conducting an ICC assay for visual confirmation, it was found that the expression of CSC marker proteins was reduced (Figure 5B). Figure 5C shows that the sphere formation ability, which is a characteristic of CSCs, was regulated by RanBP1. As a result, the sphere-forming ability of GSCs was reduced by RanBP1. The EMT phenomenon was regulated by RanBP1 in LCSCs. In order to observe whether the EMT phenomenon is regulated by RanBP1 in GSCs, the expression patterns of EMT marker proteins were examined (Figure 5D). It was found that EMT marker proteins were regulated in the group in which the expression of RanBP1 was suppressed. The results of the ICC assay showed that EMT marker proteins were regulated according to the expression of RanBP1, consistent with the results in Figure 5D (Figure 5E). Cell movement, a major feature of EMT, was measured using a Boyden chamber (Figure 5F). Cell migration and invasion were significantly reduced in the group in which the expression of RanBP1 was suppressed. When the expression of RanBP1 was suppressed using si-RNA, the expression of *IL-18* was reduced (Figure 5G).

## 3. Discussion

This study investigated the effect of a high expression of *RanBP1* on LCSCs and GSCs. RanBP1 is known to act as an adjuvant to Ran and to be involved in mitosis and in the cell cycle [30]. RanBP1 shows the highest activity in the early and late stages of the cell cycle, in order to help Ran function [31]. Additionally, RanBP1 plays an essential role in the directional migration of neural crest cells during development [32]. Several researchers have reported that RanBP1 is involved in cancer malignancy [33]. In particular, RanBP1 is known to be involved in cancer induction by regulating pre-miRNA nuclear export [34]. Further experiments will investigate the relationship with the pre-miRNA regulated by RanBP1 and analyze its function. The reason for this is that CSCs are regulated by multiple miRNAs. We showed that LCSC and GSC are regulated by RanBP1, and we speculated that miRNAs regulated by RanBP1 may act in the regulation of cancer malignancy. RanBP1 was also found to be a highly significant marker in the results of the Kaplan−Meier survival curve (Appendix A). Patients with a high expression of RanBP1 had a sharply decreased survival rate. This indicates that RanBP1 is an important therapeutic target in cancer. It seems that RanBP1 can be used both as a therapeutic target and as a novel marker, and its importance is expected to increase according to the results of future studies.

IL-18, regulated by RanBP1, was discovered 30 years ago and causes a variety of biological effects [31]. We found that IL-18 re-regulates RanBP1 to form an autocrine loop. The receptor for IL-18 is IL-18R, which consists of two subunits [35]. IL-18Ra is conjugated to IL-18 and IL-18Rb, an unconjugated signal chain [35]. We plan to investigate changes in the IL-18 receptor by RanBP1 through additional experiments in the future. We will also further investigate the signaling mechanism between RanBP1 and IL-18. It has also been reported that the microenvironment of multiple myeloma is regulated by IL-18 [20]. IL-18 is known to regulate the activity of immune cells by regulating the microenvironment [20]. Therefore, we plan to investigate the relationship between RanBP1 and immune cells, and to study whether IL-18 is involved. Known to play an important role in the response of the innate immune system, the inflammasome is responsible for the activation of inflammatory cytokines such as IL-1β and IL-18 [36]. The relationship between the inflammasome and IL-18, which is regulated by RanBP1, will be further studied. 

Our results show that RanBP1 regulates LCSC and GSC and is an important factor that can regulate EMT. In addition, IL-18 is involved in this process and regulates RanBP1 again. All these processes indicate that RanBP1 could be a valuable new therapeutic target.

## 4. Materials and Methods

### 4.1. Cell Culture and Sphere-Formation Assays

A549 (CCL-185™) and H460 (HTB-177™) human lung cancer cell lines were purchased from the ATCC and grown using RPMI 1640 MEDIUM(1X) (cat. no. SH30027.01; Hyclone; Cytiva, Pittsburgh, PA, USA). Human glioblastoma cell line U-87 MG (30014) was obtained from the Korea Cell Line Bank and cultured in DMEM/HIGH GLUCOSE (cat. no. SH30243.01; Hyclone; Cytiva, Pittsburgh, PA, USA). These media were supplemented with 10% FBS (cat. no. SH30919.03; Hyclone; Cytiva, Pittsburgh, PA, USA) and 1% Penincillin−Streptomycin Solution (cat.no. SV30010; Hyclone; Cytiva, Pittsburgh, PA, USA). All cells were cultured at a humidified incubator with 37 °C in 5% CO_2_. Tumorsphere medium (TM) was used during the sphere formation assay. TM consisted of DMEM/F12 (1:1) (cat. no. 11320-033; Invitrogen; Thermo Fisher Scientific, Inc., Waltham, MA, USA), basic fibroblast growth factor (bFGF; 20 ng/mL; cat. no. 13256-029; Invitrogen; Thermo Fisher Scientific, Inc., Waltham, MA, USA), epidermal growth factor (20 ng/mL; cat. no E9644; Sigma-Aldrich, Burlington, VT, USA), N-2 Supplement (500 µL; cat. no. 17502048; Invitrogen; Thermo Fisher Scientific, Inc., Waltham, MA, USA), and B27 serum-free supplement (1 mL; cat. no. 17504-044; Invitrogen; Thermo Fisher Scientific, Inc., Waltham, MA, USA).

### 4.2. Neutralization Assay

The CXCL1 antibody (9µg/mL; cat. no. MAB275; R&D SYSTEMS, Minneapolis, MN, USA), IL-8/CXCL8 (0.4µg/mL; cat. no. MAB208; R&D SYSTEMS, Minneapolis, MN, USA), and IL-18 (1.2µg/mL; cat. no. AF2548; R&D SYSTEMS, Minneapolis, MN, USA) were used for the neutralization assay. The normal mouse IgG1 antibody (1 µg/mL; cat. no. sc-3877; Santa Cruz Biotechnology, Inc., Dallas, TX, USA) was used as the control antibody. The experiment was conducted in the same manner as the cell culture environment, and the subsequent experiments or results were performed/obtained after 24–48 h.

### 4.3. Antibodies

Antibodies against RanBP1 (cat. no. 8780; Cell Signaling Technology, Inc., Danvers, MA, USA), CD44 (cat. no. 3570; Cell Signaling Technology, Inc., Danvers, MA, USA), Sox2 (cat. no. 3579; Cell Signaling Technology, Inc., Danvers, MA, USA), Oct-4 (cat. no. 2750; Cell Signaling Technology, Inc., Danvers, MA, USA), Nanog (cat. no. 4893; Cell Signaling Technology, Inc., Danvers, MA, USA), β-actin (cat. no. sc-47778; Santa Cruz Biotechnology, Inc., Dallas, TX, USA), ZEB1 (cat. no. sc-25388; Santa Cruz Biotechnology, Inc., Dallas, TX, USA), SLUG (cat. no. sc-166476; Santa Cruz Biotechnology, Inc., Dallas, TX, USA), SNAIL (cat. no. sc-10432; Santa Cruz Biotechnology, Inc., Dallas, TX, USA), Twist (cat. no. sc-15393; Santa Cruz Biotechnology, Inc., Dallas, TX, USA), ALDH1A1 (cat. no. ab6192; Abcam, Cambridge, UK), ALDH1A3 (cat. no. ab129815; Abcam, Cambridge, UK), E-Cadherin (cat. no. ab15148; Abcam, Cambridge, UK), CD133 (cat. no. ab19898; Abcam, Cambridge, UK), N-Cadherin (cat. no. 610920; BD Transduction, San Diego, CA, USA), Vimentin (cat. no. MA5-14564; Invitrogen, Waltham, MA, USA), CXCL1 (cat. no. PA5-115328; Invitrogen, Waltham, MA, USA), IL-8/CXCL8 (cat. no. MAB208; R&D SYSTEMS, Minneapolis, MN, USA), and IL-18 (cat. no. AF2548; R&D SYSTEMS, Minneapolis, MN, USA) were used for Western blot analysis and Immunocytochemistry assays.

### 4.4. Small Interfering RNA (siRNA) Mediated Knockdown of RanBP1

A549 and U87 cells were transfected with siRNA targeting *RanBP1*(CACUACAUCACGCCGAUGA, CAUCGGCGUGAUGUAGUG). Subsequently, 10 pmol siRNAs were transfected using Lipofectamine RNAi MAX reagent (cat. no. 13-778-150; Invitrogen; Thermo Fisher Scientific, Inc., Waltham, MA, USA). Stealth RNAi Negative Control Medium GC (cat. no. 12935-300; Invitrogen; Thermo Fisher Scientific, Inc., Waltham, MA, USA) was used as the negative control. The cells were incubated at 37 °C for at least 48 h after transfection.

### 4.5. Western Blotting

The cells were lysed in RIPA Lysis Buffer (20-188, Millipore, Burlington, VT, USA) containing phosphatase inhibitor cocktail tablets (04906837001, Roche, Basel, Switzerland) and protease inhibitor cocktail tablets (11836153001, Roche, Basel, Switzerland). The protein concentration was measured with Protein Assay Dye Reagent Concentrate (cat. no. #5000006, BIO-RAD, Hercules, CA, USA). For a Western blot analysis, equal amounts of protein were separated on 8–15% sodium dodecyl sulfate (SDS)-polyacrylmide gels, and the separated protein was transferred to Amersham™ Protran™ 0.2 µm NC (10600001, Amersham™; Cytiva, Pittsburgh, PA, USA). After blocking the transferred membrane at room temperature by using phosphate-buffered saline (PBS) buffer containing non-fat milk (10%) and Tween 20 (0.1%) for 1 h, the membranes were treated with specific antibodies overnight in a cold chamber. After washing with Tris-buffered saline (cat. no. A0027, BIO BASIC, Markham, ON, Canada), the membranes were treated with an HRP-linked secondary antibody (Anti-rabbit IgG cat. no. 7074S or Anti-mouse IgG cat. no. 7076S; Cell Signaling Technology, Inc., Danvers, MA, USA) for 2 h at room temperature, and visualized by Western Blotting Luminol Reagent (cat. no. sc-2048, Santa Cruz Biotechnology, Inc., Dallas, TX, USA).

### 4.6. RNA Isolation

A-549 and NCI-H460 cell lines were treated with 1ml of Tri Reagent (cat. no. TR118; Molecular Research Center, Inc., Cincinnati, OH, USA) for 5 min at room temperature, and 200 µL of chloroform (cat. no. 28560-0350; JUNSEI, Tokyo, Japan) was added. After inverting the samples a few times, these samples were incubated at room temperature for 2 min. The samples were centrifuged (12,000× *g*, 4 °C, 15 min) to be phased into RNAs, DNAs, and Proteins. Then, 500 µL of RNAs was separated from the sample and 500 µL of Isopropanol (cat. no. 1.09634.1011; Merck KGaA, Darmstadt, Germany) was added to extract/precipitate the RNAs. After incubated at room temperature for 5 min, the samples were centrifuged (1200× *g*, 4 °C, 8 min). Isopropanol from the samples was aspirated and the isolated RNAs were washed using 1 mL of 75% ethyl alcohol (cat. no. 000E0690; SAMCHUN PURE CHEMICAL Co., Pyeongtaek, Republic of Korea). Then, 75% ethyl alcohol was aspirated from the samples after being centrifuged (7500× *g*, 4 °C, 5 min). The isolated RNAs were dried at room temperature for 10–15 min and 30–50 L of DEPC water was added to the dried RNAs. These RNA samples were incubated in a 60 °C heat plate for 10–15 min. After heating, these samples were cooled by ice. 

### 4.7. RNA Isolation Agarose Gel Electrophoresis

A Spectrophotometer ASP-2680 (ACTGene; Piscataway, NJ, USA) was used to measure the quantity of the extracted RNA samples. First, 1µg of each RNA was added into Maxime RT Premix (Random primer) (cat. no. 25082; LiliF Diagnostics, Seongnam, Republic of Korea) to make cDNA. DEPC water was used to fulfill the total volume as 20 L. For RT-qPCR, ALDH1A1 (F: TTAGCAGGCTGCATCAAAAC, R: GCACTGGTCCAAAAATCTCC, 56 °C, 34 Cycle), ALDH1A3 (F: ACCTGGAGGGCTGTATTAGA, R: GGTTGAAGAACACTCCCTGA, 57.5 °C, 34 Cycle), CD133 (F: CATGGCCCATCGCACT, R: TCTCAAAGTATCTGG, 55 °C, 34 Cycle), RanBP1 (F: CTCCTGAAGCACAAGGAGAA, R: GTGCATTCTCAGCATTCAGG, 57 °C, 34 Cycle), GAPDH (F: AGTCAACGGATTTGGTCGTA, R: GTCATGAGTCCTTCCACGAT, 56 °C, 34 Cycle), CXCL1 (F: ATGGCCCGCGCTGCTCTCTC, R: TCAGTTGGATTTGTCACTGTTC, 56 °C, 34 Cycle), IL-8 (F: ATGACTTCCAAGCTGGCCGTG, R: TGAATTCTCAAGCCCTCTTCA, 55 °C, 35 Cycle), and IL-18 (F: GCTTGAATCTAAATTATCAGTC, R: GAAGATTCAAATTGCATCTTAT, 56 °C, 34 Cycle) primers were used. Here, 18 L of DEPC water and 2 L of each primers were added in the Maxime PCR PreMix (i-MAX II for 20 L). We used T100Thermal Cycler (Bio-Rad Laboratories, Inc., Hercules, USA) for RT-qPCR. After all of the steps, TAE buffer (cat. no. IBSD-BT002; iNtRON Biotechnology, Seongnam, Republic of Korea) and Certified Molecular Biology Agarose (cat. no. 161-3102; Bio-Rad Laboratories, Inc., Hercules, CA, USA) were mixed to make 1% gel. Then, 20 L of sample was loaded on the gels for 10–20 min. The loaded gels were put on the Desktop Gel Imaging System (ETX-20.M; EEC Biotech Co., Ltd., Bangkok, Thailand) to be pictured.

### 4.8. Immunocytochemistry

Cells were grown onto glass coverlips in 35 mm plates and fixed with 4% paraformaldehyde (cat. no. P2031; Biosesang, Seongnam, Republic of Korea) for 30 min at room temperature. After cell fixation, the cells were incubated with antibodies in a solution of Tris-buffered saline (cat. no. A0027, BIO BASIC, Markham, ON, Canada) at 4 °C for overnight. The antibodies used were RanBP1 (cat. no. 8780; Cell Signaling Technology, Inc., Danvers, MA, USA), ALDH1A1 (cat. no. ab6192; Abcam, Cambridge, UK), ALDH1A3 (cat. no. ab129815; Abcam, Cambridge, UK), CD133 (cat. no. ab19898; Abcam, Cambride, UK), CD44 (cat. no. 3570; Cell Signaling Technology, Inc., Danvers, MA, USA), E-Cadherin (cat. no. ab15148; Abcam, Cambridge, UK), N-Cadherin (cat. no. 610920; BD Transduction, San Diego, CA, USA), and Vimentin (cat. no. MA5-14564; Invitrogen, Waltham, MA, USA). Staining was visualized using Alexa Fluor 488-conjugated anti-rabbit IgG antibody (Invitrogen, Waltham, MA, USA). The nuclei were counterstained using 4,6-diamidino-2-phenylindole (DAPI; Sigma-Aldrich, Burlington, VT, USA). The stained cells were analyzed using a Zeiss LSM510 Meta microscope (Carl Zeiss Micro Imaging GmbH, Göttingen, Germany).

### 4.9. Limited Dilution Assay

In a limited dilution assay, the cells were plated in 200 μL spheroid formation assay medium in ultra-low adhesion 96-well plates. A total of 1, 10, 50, 100, or 200 cells/well were plated, with 48 wells for each starting density of cells. Oncospheres were analyzed using a light microscope (400× magnification) after 10–14 days of incubation. A well with at least one spheroid with a diameter ≥ 100 µm was defined as a positive well, and the number of positive wells was counted.

### 4.10. Invasion and Migration Assays

Migration assays were performed using an uncoated chamber (cat. no. 3422; 8-μm pore; Corning, Inc., Corning, NY, USA) and the ability of cells to migrate was measured. Invasion assays were performed by coating the chamber with Matrigel^®^ according to the manufacturer’s protocol. The lower chamber of the Transwell inserts (Cell Biolabs) was filled with 800 μL RPMI 1640 supplemented with 10% FBS. In the upper chamber, 150 μL serum-free medium (Opti-MEM^®^; cat. no. 31985-070; Invitrogen; Thermo Fisher Scientific, Inc., Waltham, MA, USA) containing 2 × 10^5^ cells was added. The cells were incubated for 24 h at 37 °C in a humidified incubator with 5% CO_2_. Cells that had migrated/invaded to the bottom of the chamber were stained with crystal violet (cat. no. HT90132-1L; Sigma-Aldrich; Merck KGaA, Burlington, VT, USA) and the cells were counted under a light microscope (400× magnification).

### 4.11. Wound Healing Assay

The cells were plated in a 60 mm culture dish and grown to 80% confluence. A wound was created by scraping the monolayer of cells with a 200 μL pipette tip in the middle. Floating cells were removed by washing with PBS and fresh medium containing 10% FBS was added. The doubling time of the A549 cells used was 24 h. The cells were incubated at 37 °C for 24 h, and imaged using phase-contrast microscopy (400× magnification). The distance between the edges of the wounds shown in the image was measured randomly at three or more places and the mean of the three measurements were obtained.

### 4.12. Colony-Formation Assay and Irradiation

Cells were seeded at a density of 1 × 10^3^ cells per 35 mm cell culture dish (cat. no. 430165; Corning, Inc., Corning, NY, USA), and then allowed to adhere for 24 h in a humidified incubator with 5% CO_2_ at 37 °C. The following day, the cells were irradiated with 3 Gy γ-radiation (KAERI). After 10–14 days, the cells were stained for colonies (defined as clusters of ≥50 cells) with 0.5% crystal violet for 1h at room temperature, and the stained colonies were counted. Clonal survival rates were expressed as a percentage of the non-irradiated control group.

### 4.13. Cytokine Array

The Human Cytokine Array Kit (cat. no. ARY005B; R&D SYSTEMS, Minneapolis, MN, USA) was used to evaluate the secreted factors regulated by RanBP1 in lung cancer stem cells. The A549 cell was transfected with siRNA targeting *RanBP1*. All of the reagents were brought to room temperature before use. Here, 2 mL of Block Buffer was pipetted into each well of the four-well multi-dish and incubated for 1 h on a rocking platform shaker. While the arrays were blocking, the samples were prepared by adding up 1ml of each sample to 0.5 mL of Block Buffer. Then, 15 µL of reconstituted Human Cytokine Array Detection Antibody Cocktail was added to each prepared sample and it was incubated at room temperature for 1 h. After 1 h, Block Buffer was aspirated from the four-well multi-dish, and then the prepared samples were added. The samples were then incubated overnight at 2–8 °C on a rocking platform shaker. Each membrane was carefully removed from the four-well multi-dish and placed into plastic containers with 20 mL of Wash Buffer. Each membrane was washed with 1×Wash Buffer for 10 min on a rocking platform shaker three times. Then, 2 mL of diluted Streptavidin-HRP were pipetted into each well of the four-well multi-dish. After 30 min of incubation on a rocking platform shaker, each membrane was carefully removed from the four-well multi-dish and placed into plastic containers with 20 mL of Wash Buffer. Each membrane was washed with Wash Buffer for 10 min on a rocking platform shaker three times. After washing, each membrane was removed from the container and visualized by Chemi Reagent Mix.

### 4.14. Kaplan−Meier Plotter

Using a published genetic information system, Kaplan−Meier survival values were obtained (kmplot.com/analysis). This was based on the results of mRNA gene chip analysis using the tissues from lung cancer patients. The gene symbol used was *RanBP1*. All of the conditions were set as the total lung cancer patients.

### 4.15. Statistical Analysis

All of the experiments were performed by repeating at least three independent experiments, and the results are expressed as the mean ± standard deviation. Each exact n value is displayed in the corresponding figure legend. To validate the data, all graphs were compared using a two-sided paired Student’s *t*-test. *p* < 0.05 was considered to indicate a statistically significant difference.

## 5. Conclusions

The *RanBP1* gene is overexpressed in ALDH1^+^ lung cancer cells and CD133^+^ glioma cells.The overexpression of RanBP1 regulated the marker proteins and regulatory proteins of lung cancer stem cells and glioma stem cells, and modulated the ability of lung cancer cells to resist irradiation.RanBP1 regulated EMT marker proteins and regulatory proteins, and modulated cell migration and invasion ability.The cytokine IL-18 was regulated by RanBP1, and IL18 regulated cancer stem cell and EMT characteristics.

## Figures and Tables

**Figure 1 ijms-24-06855-f001:**
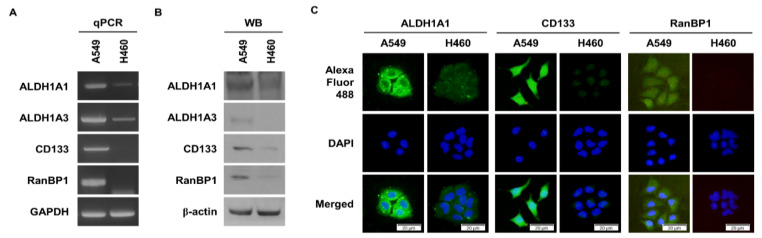
High expression of RanBP1 in A549 cells. (**A**) Comparison of the gene expression of CSC markers in A549 and H460 cells. GAPDH is used as a loading control. (**B**) Comparison of the expression of CSC marker proteins in A549 cells and H460 cells. β-actin is used as a loading control. WB: Western blotting. (**C**) Comparison of CSC marker protein expression through an ICC assay.

**Figure 2 ijms-24-06855-f002:**
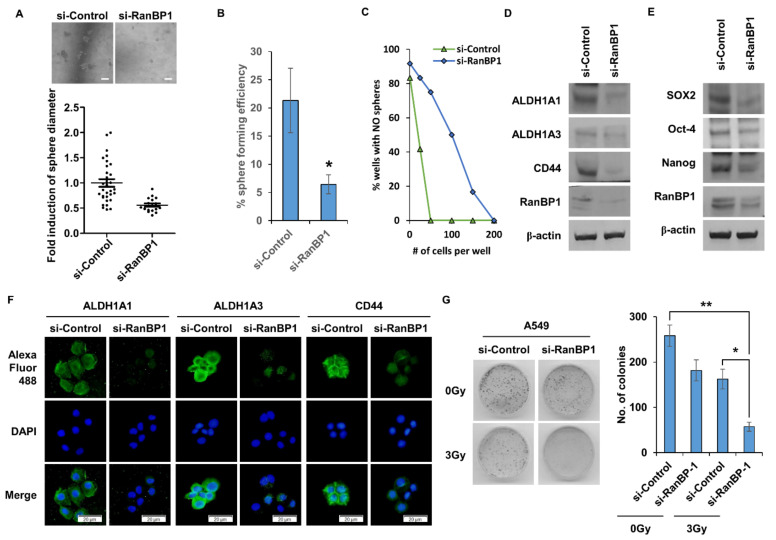
RanBP1 regulates the properties of CSCs in lung cancer cells. (**A**) A549 cells are transfected with si-RNA targeting RanBP1, and the sphere formation ability is analyzed. (**B**) *RanBP1* gene inhibition using si-RNA followed by single cell assay. Experiments are performed in duplicate five times. (**C**) Limiting dilution assay is performed on 96-well plates. Here, 1, 50, 100, 150, and 200 cells per well are seeded. Results are confirmed 10 days after cell seeding. (**D**) Expression analysis of CSC marker proteins ALDH1A1, ALDH1A3, and CD44 according to the expression of RanBP1. (**E**) Western blot expression analysis of the CSC regulatory proteins SOX2, Oct-4, and Nanog. A549 cells are transfected with si-RNA targeting RanBP1. (**F**) Immunocytochemical analysis of CSC marker proteins using si-RanBP1-treated A549 cells. An antibody against the CSC marker protein is used as the primary antibody, and an antibody labeled with Alexa Fluor 488 is used as the secondary antibody. (**G**) Colony formation assay to view clonogenesis with A549 cells transfected with siRNA. Cells are irradiated with a 3Gy dose after 24 h. After 10 days of culture, colonies are stained with crystal violet. Error bars represent mean ± SD. Triplicate samples. * *p* < 0.05, ** *p* < 0.0005 versus control. Scale bar = 50 μm.

**Figure 3 ijms-24-06855-f003:**
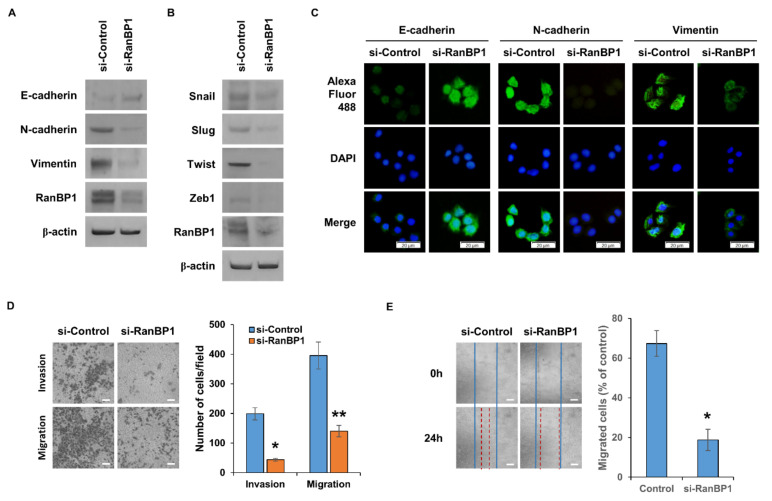
EMT is regulated by RanBP1 in lung cancer cells. (**A**) Western blot analysis of EMT marker proteins E-cadherin, N-cadherin, and Vimentin. A549 cells are transfected with si-RNA targeting RanBP1. (**B**) The EMT regulatory proteins Snail, Slug, TWIST, and ZEB1 are also analyzed by Western blot. The expression of RanBP1 is suppressed using si-RNA. (**C**) Immunocytochemical analysis of EMT marker proteins after transfection with si-RanBP1 in A549 cells. Primary antibodies targeting each marker protein are used, and secondary antibodies are labeled with Alexa Fluor 488. (**D**) Migration and invasion assay of cells by si-RNA transfected A549 cells. (**E**) Wound healing assay using RanBP1 knockdown A549 cells. Error bars represent mean ± SD. Triplicate samples. * *p* < 0.0005, ** *p* < 0.001 versus control. Scale bar = 50 μm.

**Figure 4 ijms-24-06855-f004:**
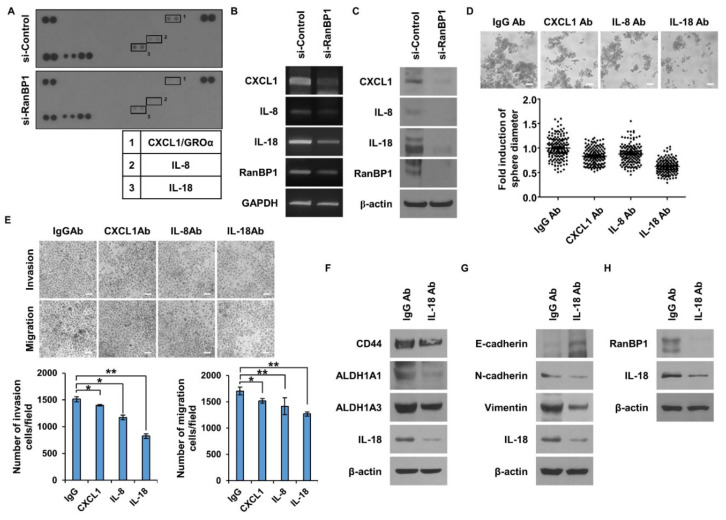
Cytokines regulated by *RanBP1* in lung cancer. (**A**) Identification of secreted factors regulated by *RanBP1* using cytokine arrays. A549 cells are used, and the expression of *RanBP1* is suppressed using si-RNA. (**B**) Comparison of the gene expression of cytokines regulated by RanBP1. (**C**) Analysis of cytokines regulated by RanBP1 using WB. (**D**) Comparison of the sphere formation ability after each neutralizing antibody treatment. The p value is less than 0.0001 versus the IgG Ab value. (**E**) Comparison of migration and invasion ability of cells after each neutralizing antibody treatment. Experiments performed in triplicate. (**F**) CSC marker protein analysis in A549 after treatment with IL-18 neutralizing antibody. (**G**) Confirmation of the expression levels of EMT marker proteins after treatment with the IL-18 neutralizing antibody. (**H**) Expression analysis of RanBP1 regulated by IL-18 using neutralizing antibodies. Error bars represent mean ± SD. Triplicate samples. * *p* < 0.05, ** *p* < 0.001, versus control. Scale bar = 50 μm.

**Figure 5 ijms-24-06855-f005:**
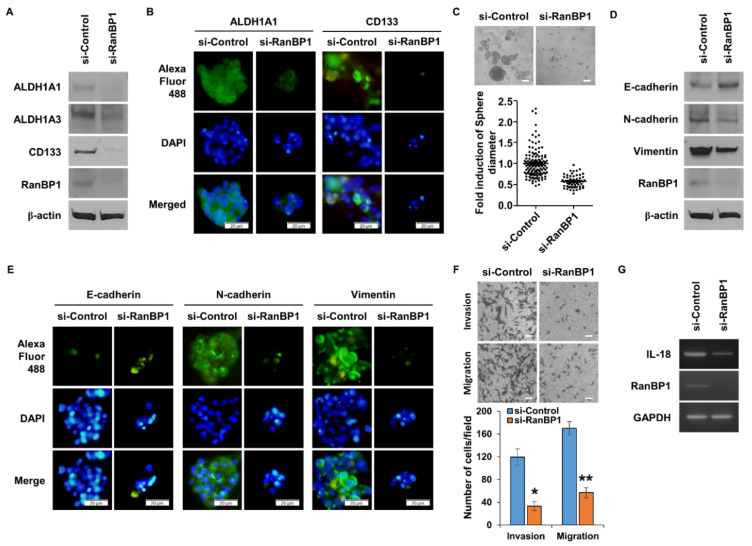
Characteristics of CSC and EMT events regulated by *RanBP1* in glioma. (**A**) Comparison of the expression of GSC marker proteins by *RanBP1* gene suppression in GSCs. U87 cells are differentiated into GSCs using TM. (**B**) Comparison of the marker protein expression according to the expression of RanBP1 using ICC in GSCs. (**C**) Comparison of differences in sphere formation according to the expression level of RanBP1 in GSCs. Experiments performed in triplicate. (**D**) Comparison of the EMT marker protein expression after *RanBP1* gene suppression using si-RNA. (**E**) Confirmation of EMT marker protein expression according to the expression of RanBP1 using ICC. (**F**) Analysis of cell migration and invasion ability after the suppression of RanBP1 expression using si-RNA. (**G**) Analysis of regulation of *IL-18* expression by RanBP1 in GSCs. Error bars represent mean ± SD. Triplicate samples. * *p* < 0.001, ** *p* < 0.0005, versus control. Scale bar = 50 μm.

## Data Availability

Not applicable.

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
