# Peer review of "RanBP1: A Potential Therapeutic Target for Cancer Stem Cells in Lung Cancer and Glioma"

_ijms, 2023, doi:10.3390/ijms24076855_

Round 1

Author Response

We sincerely appreciate your interest in and review of our paper. We've made revisions and improvements based on your suggestions. We've provided explanations and fixes for your suggestions in six areas. We've also fixed all inaccuracies and typos.

  • Please see the attachment for more details.

Reviewer 2 Report

Yeon-Jee Kahm and colleagues described in this manuscript the RanBP1 mediated secretion of IL-18 regulates lung cancer stem cells (LCSCs) and glioma stem cells (GSCs). They analyzed the RanBP1 overexpression and association with poor patient prognosis. They also reported that RanBP1 regulates the ability to resist radiation of cancer stem cells, the phenomenon of the metastasis-associated epithelial-mesenchymal transition. RanBP1 modulates the cancer microenvironment by regulating IL-18, a cytokine associated with cancer malignancy. Their endpoint conclusion is that the RanBP1 in LCSCs and GSCs is a potential therapeutic target. Although the manuscript generally presented in a logical order, the text should be revised with more attention to improve clarity of each section especially discussion. I believe at this stage, this manuscript not able to move forward, but after substantial input from the authors may considered for publication in The International Journal of Molecular Sciences, MDPI. The authors should consider the following major and some minor issues to improve the strength of this manuscript:   

1.    Can authors explain why the RanBP1 protein expression pattern was not similar in all the experiments. Some of the figures included single, but others are double bands. Also, the source of this antibody (cat. no. 9255; Cell Signaling Technology, Inc.) mentioned in lines 81 & 120 has shown completely different from company website (https://www.cellsignal.com/product/productDetail.jsp?productId=9255) that is double bands Phospho-SAPK/JNK (Thr183/Tyr185) (G9) Mouse mAb #9255. Only one single band RanBP1 antibody (cat. No. 8780) is available in Cell Signaling Technology, Inc. that is mentioned for only western blot application (https://www.cellsignal.com/products/primary-antibodies/ranbp1-antibody/8780?N=2189226546+4294967292+4294956287&Nrpp=60&No=240&fromPage=plp).  

2.    I am not sure in what basis the authors reporting here RanBP1 is highly expressed in A549 cells. Published paper (cited by the authors in this manuscript; doi: 10.1158/1541-7786.MCR-08-0393) reported that H82 and H1299 are negative, but H460 is ALDH1-positive. As ALDH1 is a tumor stem cell-associated marker in lung cancer, how the authors compared the expression of RanBP1 in A549 vs. H460 (both the cell lines are acquired lung cancer stem cell like properties). 

3.    I am not satisfied with the efficiency of RanBP1 silencing using siRNA in figure 4B as the reduction of this gene expression is negligible in si-RanBP1 compared to si-control. Therefore, I would like to ask the authors to provide the protein expression for the proof of RanBP1 silencing in figure 4. 

4.    I didn’t find any figure related for the statement (line 329) of “RanBP1 is a gene highly expressed in GSCs”.

5.    IL-1β and IL-18 are closely related and activated mostly through inflammasomes, an emergent topic in lung cancer. As the authors reported in this manuscript that and IL-18 is regulated by RanBP1 and associated with malignancy of lung cancer cells. Readers may be curious to know about the IL-1β under these circumstances.

6.    Need to be mentioned sources of all the antibodies used for western blots in “Materials and Methods” section. 

7.    Spelling mistake in line 18: “phemomenom” would be “phenomenon”.

8.    What the authors mean “CM medium”. Need to be define completely.

9.    For neutralization assay, need complete information in “Materials and Methods” section about the concentration of antibodies used for blocking.

10. I will recommend the authors to include more details in each “Figure legends”., Especially, how many biological or experimental replicates were performed by them. 

Author Response

Thank you for your interest in our paper and your advice. We've taken all of our editors' suggestions and incorporated them into the paper. You can find our responses to the suggestions and explanations of the changes in the attachments below.

Round 2

Reviewer 1 Report

The current version of the manuscript is noticeably improved, but it still has flaws that need to be corrected.

Line 63.  Instead of “These medium were supplemented...” should be “These media were supplemented…”

Line 65-66, 259. A conditioned medium is a medium enriched with the secretome of some cell culture. Adding recombinant factors to a medium does not make it “conditioned”. The term should be changed.

Line 203-205. As I already wrote in my first review, conventional PCR cannot be used to assess the level of gene expression. Quantitative real-time PCR was to be used. In this regard, the level of expression should not be mentioned in the context of Figure 1A. You can only indicate the presence or absence of the expression. Comparison of the expression level is better to place in the context of Figure 1B, since WB is a semi-quantitative method.

Figure 1 A-B. The methods used to obtain the blots should be noted in the figure captions.

Line 248-251, Figure 1 D. I still think that survival analysis and figure 1D is not in the right place. It is better to make a separate small figure in the discussion section.

Line 329. The title of the section is not clear

Author Response

Thank you for your quick response, and most importantly, thank you for your interest in our paper. We have revised the paper according to the reviewer's suggestions. We have provided a brief response to each suggestion in the text below. 

The current version of the manuscript is noticeably improved, but it still has flaws that need to be corrected.

Line 63.  Instead of “These medium were supplemented...” should be “These media were supplemented…”

-> We have modified the relevant sentence as suggested by the reviewer.

Line 65-66, 259. A conditioned medium is a medium enriched with the secretome of some cell culture. Adding recombinant factors to a medium does not make it “conditioned”. The term should be changed.

-> Changed from "Conditioned medium" to "Tumorsphere medium".

Line 203-205. As I already wrote in my first review, conventional PCR cannot be used to assess the level of gene expression. Quantitative real-time PCR was to be used. In this regard, the level of expression should not be mentioned in the context of Figure 1A. You can only indicate the presence or absence of the expression. Comparison of the expression level is better to place in the context of Figure 1B, since WB is a semi-quantitative method.

-> Corrected the sentence in Figure 1A that mentions expression levels.

Figure 1 A-B. The methods used to obtain the blots should be noted in the figure captions.

-> We have added qPCR and WB.

Line 248-251, Figure 1 D. I still think that survival analysis and figure 1D is not in the right place. It is better to make a separate small figure in the discussion section.

-> We have moved the relevant figures to Supplementary figures.

Line 329. The title of the section is not clear

-> We have corrected the title of the section.

Reviewer 2 Report

I will recommend the authors carefully revise manuscript with the corresponding reference(s) again before the final submission.

Also, the authors need to be replaced the source of RanBP1 antibody, cat. no. 9255; Cell Signaling Technology, Inc. mentioned specially in line 159 by cat. no. 8780; Cell Signaling Technology, Inc.

Author Response

I will recommend the authors carefully revise manuscript with the corresponding reference(s) again before the final submission.

-> Thank you for your interest in our paper. The reviewer's suggestions have made the paper even better. We have taken note of the suggestions and carefully revised the manuscript.

Also, the authors need to be replaced the source of RanBP1 antibody, cat. no. 9255; Cell Signaling Technology, Inc. mentioned specially in line 159 by cat. no. 8780; Cell Signaling Technology, Inc.

-> We have made the relevant corrections.